# Cognitive Training Improves Joint Stiffness Regulation and Function in ACLR Patients Compared to Healthy Controls

**DOI:** 10.3390/healthcare11131875

**Published:** 2023-06-28

**Authors:** Yong Woo An, Kyung-Min Kim, Andrea DiTrani Lobacz, Jochen Baumeister, Jill S. Higginson, Jeffrey Rosen, Charles Buz Swanik

**Affiliations:** 1Department of Health and Human Sciences, Loyola Marymount University, Los Angeles, CA 90621, USA; 2Department of Sport Science, Sungkyunkwan University, Suwon-si 16419, Republic of Korea; km.kim@g.skku.edu; 3Department of Kinesiology and Sport Sciences, University of Miami, Coral Gables, FL 33146, USA; 4Nursing and Health Sciences, Neumann University, Aston, PA 19014, USA; lobacza@neumann.edu; 5Department of Exercise & Health, Paderborn University, 33098 Paderborn, Germany; jochen.baumeister@uni-paderborn.de; 6Department of Mechanical Engineering, University of Delaware, Newark, DE 19716, USA; higginso@udel.edu; 7Department of Psychological and Brain Sciences, University of Delaware, Newark, DE 19716, USA; jrosen@psych.udel.edu; 8Department of Kinesiology and Applied Physiology, University of Delaware, Newark, DE 19716, USA; cswanik@udel.edu

**Keywords:** neuroplasticity, neuromuscular control, kinesiophobia, functional joint instability, neurocognition

## Abstract

As cognitive function is critical for muscle coordination, cognitive training may also improve neuromuscular control strategy and knee function following an anterior cruciate ligament reconstruction (ACLR). The purpose of this case-control study was to examine the effects of cognitive training on joint stiffness regulation in response to negative visual stimuli and knee function following ACLR. A total of 20 ACLR patients and 20 healthy controls received four weeks of online cognitive training. Executive function, joint stiffness in response to emotionally evocative visual stimuli (neutral, fearful, knee injury related), and knee function outcomes before and after the intervention were compared. Both groups improved executive function following the intervention (*p* = 0.005). The ACLR group had greater mid-range stiffness in response to fearful (*p* = 0.024) and injury-related pictures (*p* = 0.017) than neutral contents before the intervention, while no post-intervention stiffness differences were observed among picture types. The ACLR group showed better single-legged hop for distance after cognitive training (*p* = 0.047), while the healthy group demonstrated no improvement. Cognitive training enhanced executive function, which may reduce joint stiffness dysregulation in response to emotionally arousing images and improve knee function in ACLR patients, presumably by facilitating neural processing necessary for neuromuscular control.

## 1. Introduction

Emerging research suggests that between psychological factors, specifically, a heightened fear of re-injury, play a significant role in diminished knee function along with a lower success rate of returning to pre-injury levels of physical activity following an anterior cruciate ligament (ACL) injury [1,2,3,4]. It has been found that negative emotions have the potential to disrupt existing neural action-planning networks in the central nervous system (CNS) [5], highlighting the need for advanced cognitive management strategies. These strategies also are essential for maintaining appropriate motor control and safeguarding the joint in the face of unforeseen events, particularly competitive athletic maneuvers [6,7]. It has been extensively emphasized the pivotal role of cognitive skills in maintaining muscle coordination, which involves monitoring environmental cues and concurrently adjusting movement planning [8]. One prospective study by Swanik et al. [7] found that a group of collegiate athletes who had lower performance on neurocognitive tests, including slower processing speed, reactions, poorer memory, and visual–spatial abilities, went on to suffer noncontact ACL sprains.

It has been established that cognitive training can improve the proficiency of neural processing in the frontal areas [6,9,10], such that enhanced cognitive processing may quickly suppress cortical responses in the prefrontal cortex to regulate emotions. This fear network in the prefrontal cortex is highly associated with cognitive-action-planning circuits [6,9,10]. Both cognitive loading and startle responses lead to stiffness dysregulation, which coincides with the rapid events of unintentional noncontact ACL injuries [11,12,13]. Therefore, improving executive function, which is the ability to cognitively control behaviors, could provide sufficient cognitive processing necessary for anticipatory motor programming and maintaining functional joint stability [7]. However, limited research exists for evaluating the effect of cognitive training on muscle stiffness regulation strategies and function following ACL reconstruction (ACLR). The absence of these data has created a barrier to our understanding of joint instability in ACLR patients, as well as best practices to maximize the functional outcomes of each patient. Therefore, the purpose of this study was to examine the effects of cognitive training on joint stiffness regulation in response to negative visual stimuli and knee function following ACLR compared to healthy individuals. We hypothesize that cognitive training may improve joint stiffness regulation strategies and knee functional outcomes.

## 2. Materials and Methods

### 2.1. Participants

Forty volunteers (twenty ACLR patients, twenty healthy controls) between the ages of 18 and 45 who were recruited from within and outside University and a local orthopedic clinic over 6 months (November 2015–April 2016) participated in a case-control study. Participant demographics are presented in Table 1.

ACLR participants (ACLR) were included if they had a surgical repair for unilateral ACL rupture (time from surgery, 3.9 ± 2.2 years), but cleared to return to pre-injury levels of physical activity by their physician. Healthy controls (CONT), who were matched to by age, gender, and leg dominance, regularly maintained moderate physical activity levels at least three days/week. Following a pretest session, all participants were assigned to a 4-week cognitive training program (Figure 1).

Measures of knee function outcomes, executive function skills, and joint stiffness in response to emotionally evocative pictures were conducted before and after 4 weeks of cognitive training. PRE—before the cognitive training; POST—after the cognitive training; KOS-ADL—knee outcome survey-activities of daily living; GRKF—global rating of knee function; NIH-TB—NIH Toolbox for executive function assessment; FICA—flanker inhibitory control and attention test; DCCS—dimensional change card sort test; SPAD—a custom-built stiffness and proprioception assessment device; IAPS—international affective picture system; QMVIC—quadriceps maximum voluntary isometric contraction; HMVIC—hamstring maximum voluntary isometric contraction.

Individuals with a recent history of lower extremity injury or surgery, within the last six months, neurological issues, or hearing impairments that could potentially affect the assessment of executive function, joint stiffness, and/or knee functional outcomes were not included as participants in the study. Participants who did not fulfill the requirement of completing at least 10 h of cognitive training [14], experienced a new lower extremity injury during the intervention, or failed to attend the post-test session were also excluded from the study. Consequently, three individuals from the CONT group and four from the ACLR group were excluded due to incomplete cognitive training. Prior to the pretest session, all participants received and signed an institution-approved informed consent form. The number of participants per group was calculated by using a priori power analysis using 0.25 for an effect size f, 80% of power with a 0.05 type I error rate, and we had 12 participants per group.

### 2.2. Protocol and Signal Processing

The brain training for high achievers course offered by BrainHQ (Posit Science Corp., San Francisco, CA, USA) was utilized after the pretest session to exercise cognitive skills for at least 10 h in a 4-week period at a self-selected pace. The brain training for high achievers course is specifically designed to cater to individuals who aspire to improve cognitive performance. This comprehensive program consists of 12 computational brain exercise games that strategically target executive function skills. These games place particular emphasis on enhancing attention, brain speed, working memory, fluid intelligence, and social cognition. By addressing these cognitive domains, the course aims to optimize cognitive abilities and empower individuals to perform at their cognitive best.

The assessment of executive function performance utilized the executive function assessment tool offered by the National Institutes of Health Toolbox (NIH-TB). This tool comprised two computer-based tests: the Dimensional Change Card Sort (DCCS) and Flanker Inhibitory Control and Attention (FICA) tests. These computer-based tests are comparable to traditional paper-and-pencil cognition tests frequently used in clinical and research environments to evaluate skills related to decision making and inhibition [15,16]. Prior to each executive function test, participants were given oral instructions by the investigator and practiced for several minutes until comfortable with the tests. The order between these two tests was randomized. The progression of executive function was assessed by comparing the computed scores from 0 to 10 between pre-training and post-training for each DCCS and FICA test [15,16].

To assess joint stiffness regulation strategies, a custom-built stiffness and proprioception assessment device (SPAD) was utilized, operating at a frequency of 2400 Hz. The joint stiffness followed a specific sequence: participants performed three maximum voluntary isometric contractions (MVIC) for both the quadriceps and hamstrings. This was followed by resistance to a rapid perturbation during maximum knee extension, involving a 100°/s angular velocity and 1000°/s^2^ angular acceleration, within a 40-degree flexion arc (from 30-degree to 70-degree knee flexion). During the joint stiffness task, targeted emotions were induced using a set of 62 preselected neutral pictures (valence: 4.03–5.20, arousal: 1.72–3.46) and 60 fear-related pictures (valence: 1.31–4.32, arousal: 5.9–7.15) obtained from the International Affective Picture System (IAPS) [17]. The neutral pictures consisted of general objects such as flowers or office supplies, while the fear-related pictures included content designed to evoke fear, such as threatening images involving humans, animals, or accidents. Additionally, 60 knee-injury-related pictures, including both contact and noncontact ACL mechanisms, were included to investigate the potential impact of sports-specific stimuli on joint stiffness. Sports-type images were selected according to the highest ACL incidence rates: basketball, cycling, football, gymnastic, handball, soccer, ski, tennis, and wrestling [18]. An acoustic startle stimulus for each picture category was applied during presentation of one randomized picture set that contained an equal distribution of 30 neutral, fearful, and injury-related pictures [13]. A single trial was composed of 6 s of the initial black screen, a 6 s picture presentation, and a 3 s black screen.

Prior to the perturbation, randomly ordered pictures were displayed on the monitor for 800 ms. A high-pitched (1000 Hz) noise exceeding 100 dB sound pressure level, lasting 10 ms, was delivered through headphones 100 ms before the perturbation to induce an acoustic startle [13]. For non-stiffness trials, different pictures were displayed for 6 s without any perturbation. The participants remained unaware of the order and quantity of trials. Raw torque and position data collected by the SPAD underwent preprocessing, which involved band-pass filtering between 20–400 Hz, rectification, and subsequent low-pass filtering at 5 Hz. The resulting smoothed torque and position data were used to calculate joint stiffness values by dividing the change in torque (Newton meter) by the change in displacement (degrees), which was then adjusted for gravity and normalized to each participant’s body weight (Nm/°/kg). The short- (0–4°), mid- (0–20°), and long-range (0–40°) stiffness values during knee flexion perturbations were reported for further analysis [13].

To assess knee function outcomes, two subjective surveys were employed. The surveys included the Knee Outcome Survey–Activities of Daily Living (KOS-ADL), which is a self-reported questionnaire measuring symptoms and functional deficits associated with knee injuries, and a visual analog scale known as the global rating of knee function (GRKF) [19,20]. The cognitive training’s impact on knee function was evaluated by calculating the percentage difference between pre-training and post-training scores for the KOS-ADL and GRKF. Additionally, the single-legged hop for distance was conducted to assess the discrepancy in knee function between limbs [21]. In this test, the limb symmetry index (LSI) was calculated as a percentage, comparing the performance of the reconstructed knee to the healthy limb in individuals with ACLR. For the CONT group, LSI was determined by comparing the performance differences between the matched knee to the ACLR group and the other limb [21].

### 2.3. Statistical Analysis

Executive function and knee functional outcomes were compared with separate 2-way repeated measures of ANOVAs with one within-subject factor (time: PRE, POST) and one between-subject factor (group: ACLR, CONT) to determine between and within-group differences. Stiffness between picture types was assessed using separate 3-way repeated measure ANOVAs with two within-subject factors (type: NEU, FEAR, INJ; time: PRE, POST) and one between-subject factor (group: ACLR, CONT) for each dependent variable. Additionally, separate 2-way repeated ANOVAs with two within-subject factors (time: PRE, POST; type: NEU, FEAR, INJ) were used to determine within-group differences for stiffness. Pairwise comparison analysis was performed when a significant interaction effect was observed. Descriptive analysis was used to identify any outliers or irregularities in the distribution. Normality of the data was assessed using the Shapiro–Wilk test. Statistical significance was set at an alpha level of 0.05. Effect sizes using partial eta-squared (η^2^) and Cohen’s d, and mean differences and associated 95% confidence intervals (CIs) were also reported to determine clinical implication on findings. Effect sizes were interpreted as small (η^2^ = 0.01, d = 0.2), medium (η^2^ = 0.06, d = 0.5), and large (η^2^ = 0.14, d = 0.8) [22].

## 3. Results

The results showed a significant group main effect for DCCS (F_[1,30]_ = 6.14, *p* = 0.02, effect size (h^2^ = 0.17)) and time main effect for FICA (F_[1,31]_ = 9.23, *p* < 0.01, effect size (h^2^ = 0.23)) tests (Table 2). Pairwise comparisons showed the ACLR had better DCCS scores than the CONT regardless of the cognitive training (PRE: *p* = 0.01, mean difference: 0.36, 95% CI: [0.08, 0.64], effect size (d = 0.93); POST: *p* = 0.01, mean difference: 0.26, 95% CI: [0.06, 0.46], effect size (d = 0.96)). Both groups improved executive function scores for the FICA test following the cognitive training (mean difference: 0.13, 95% CI: [0.04, 0.22], effect size (d = 0.50)).

A significant group-by-type interaction effect was observed for short-range stiffness (F_[1.732,38.110]_ = 5.54, *p* = 0.01, effect size (h^2^ = 0.20)) (Table 3). Pairwise comparisons revealed that the CONT produced greater short-range stiffness in response to FEAR (*p* = 0.01, mean difference: 0.006, 95% CI: [0.001, 0.010], effect size (d = 0.63)) and INJ (*p* = 0.03, mean difference: 0.006, 95% CI: [0.001, 0.011], effect size (d = 0.54)) compared to the ACLR. For mid-range stiffness, a significant main effect for type was observed (F_[2,50]_ = 6.50, *p* < 0.01, effect size (h^2^ = 0.21)). Pairwise comparisons revealed greater mid-range stiffness in response to FEAR than NEU (*p* < 0.01, mean difference: 0.009, 95% CI: [0.003, 0.014], effect size (d = 0.53)). Additionally, a significant time-by-type interaction effect for mid-range (0 to 20°) was observed in the ACLR (F_[1.415,16.986]_ = 4.91, *p* = 0.03, effect size (h^2^ = 0.29)), but not in the CONT (*p* > 0.05) (Table 3, Figure 2). Pairwise comparisons revealed that the ACLR produced greater mid-range stiffness in response to both FEAR (*p* = 0.02, mean difference: 0.010, 95% CI [0.001, 0.020], effect size (d = 0.56)) and INJ(*p* = 0.02, mean difference: 0.013, 95% CI [0.003, 0.020], effect size (d = 0.65)) than NEU before the cognitive training, while there were no post-intervention stiffness differences among emotion types (FEAR: *p* = 0.66, mean difference: 0.002, 95% CI [−0.011, 0.015], effect size (d = 0.12); INJ: *p* = 0.38, mean difference: −0.003, 95% CI [−0.015, 0.008], effect size (d = −0.25)).

The ACLR produced greater mid-range stiffness in response to fearful and injury-related pictures than neutral pictures before the cognitive training (*p* = 0.02 and *p* = 0.02, respectively), while there were no post-intervention stiffness differences among emotion types (*p* > 0.05). Abbreviations: CONT—healthy controls; ACLR—ACL reconstructed patients; NEU—neutral pictures; FEAR—fearful pictures; INJ—sports knee-injury-related pictures; PRE—before the cognitive training; POST—after the cognitive training. * Significant difference from NEU (*p* < 0.05).

A significant time-by-group interaction effect for the LSI was observed (F_[1,27]_ = 4.32, *p* < 0.05, effect size (h^2^ = 0.14)) (Table 2). Pairwise comparisons revealed that the ACLR had significantly lower functional performance in the injured knee than the other limb when compared to those in the CONT regardless of the cognitive training (PRE: *p* < 0.01, mean difference: −5.87, 95% CI [−9.76, −1.98], effect size (d = −0.99); POST: *p* = 0.02, mean difference: −3.92, 95% CI [−7.25, −0.59], effect size (d = −0.86)). However, the ACLR improved the involved limb’s hop distance after the cognitive training (*p* = 0.02, mean difference: 3.51, 95% CI [0.53, 6.48], effect size (d = 0.45)), while the CONT showed no LSI differences between before and after the cognitive training (*p* > 0.05). A significant main effect for the group was observed for GRKF (F_[1,26]_ = 5.71, *p* = 0.02, effect size (h^2^ = 0.18)), KOS-ADL (F_[1,29]_ = 8.63, *p* < 0.01, effect size (h^2^ = 0.23)), and the number of knee-giving-way episodes (F_[1,28]_ = 16.05, *p* < 0.01, effect size (h^2^ = 0.36)) (Table 2). Pairwise comparisons revealed that the ACLR had lower GRKF (PRE: *p* = 0.04, mean difference: −4.87, 95% CI [−9.62, −0.12], effect size (d = −0.81); POST: *p* = 0.02, mean difference: −3.92, 95% CI [−7.25, −0.59], effect size (d = −0.86)) and KOS-ADL (PRE: *p* < 0.01, mean difference: −6.50, 95% CI [−10.53, −2.47], effect size (d = −1.06); POST: *p* = 0.01, mean difference: −4.19, 95% CI [−7.13, −1.25], effect size (d = −1.05)) scores and a higher number of knee-giving-way episodes (PRE: *p* < 0.01, mean difference: 2.85, 95% CI [1.22, 4.48], effect size (d = 1.32); POST: *p* < 0.01, mean difference: 1.27, 95% CI [0.58, 2.50], effect size (d = 1.21)) than CONT, regardless of the cognitive training.

## 4. Discussion

In our study, both the ACLR and CONT groups demonstrated improved executive function following cognitive training. These findings align with research conducted by Ball et al. [23], who reported enhanced executive function in individuals who underwent computer-based speed-of-processing training, leading to improved accuracy and speed in visually identifying specific information. Our results from the NIH-TB executive function assessment support these findings, suggesting that cognitive computer training can enhance executive functioning skills. These improved skills may be beneficial for anticipating and responding to unforeseen threats to the knee joint during high-velocity physical maneuvers.

To replicate sudden, unanticipated joint loading, an acoustic startle event was utilized prior to a 40-degree knee flexion perturbation in our study [13]. This type of loading is the most common mechanism for noncontact ACL injury injuries [24]. DeAngelis et al. [13] conducted a study that demonstrated interrupted joint stiffness values following a startle response in healthy individuals. These alterations in joint stiffness regulation indicate a compromised mechanism for preparing and reacting to dynamic restraints, which consequently exposes the knee joint to excessive loads. This series of events suggests a disconnection between the joint structure and the CNS, indicating insufficient neuromuscular control and dysregulation of joint stiffness [13]. Our data indicated that the ACLR group exhibited lower short-range stiffness values compared to the CONT group when participants were exposed to both general fearful and specific injury-related pictures before the knee perturbation with the acoustic startle event. Short-range stiffness reflects the rapid resistance to sudden joint perturbations provided by the passive viscoelastic properties of connective tissues, as well as the reverse pivoting and existing actin-myosin cross-bridges within muscles [25]. This involuntary muscle resistance is thought to be greatly associated by the fusimotor muscle spindle system, which determines the amount of resting muscle stiffness or tone [26]. Potent negative emotional responses could lead to sudden changes in muscle tone [27] because parasympathetic dominant neurophysiological emotional responses may decrease sensitivity of the muscle spindle system [28]. Decreased short-range stiffness in the ACLR may be indicative of the altered neuromechanical coupling strategy between joint structures and the CNS, possibly due to diminished neural sensitivity within the muscle spindle system resulting from negative stimuli [26,28]. As a result, individuals who have undergone ACLR may encounter challenges in initially augmenting the stiffness of the knee joint structure via the fusimotor spindle system when reacting to abrupt fearful stimuli. This suggests that improved dynamic restraint mechanisms are necessary to compensate for deficits in involuntary muscle stiffness and maintain functional joint stability.

The findings of our study demonstrate that the cognitive training led to improved strategies for regulating mid-range joint stiffness in the ACLR group, particularly when faced with negative emotional stimuli during an unexpected knee perturbation accompanied by a startle event. The mid-range stiffness encompasses both passive and dynamic restraint components of the knee’s surrounding muscles [11,26]. While the knee is loaded, the CNS must precisely interpret sensory feedback from the thigh muscles with respect to instantaneous changes in force, length, and joint position, in order to continuously regulate optimal joint stiffness [24]. Schmitz and Shultz [29] investigated joint stiffness and muscle absorption during drop jumps and showed that lower joint stiffness throughout the entire landing period was correlated with greater force absorption in the knee muscles. Swanik et al. [24] also observed that ACL patients with relatively normal knee function, in comparison to healthy controls, exhibited reduced muscle stiffness with heightened activation of the hamstring muscles. This decrease in stiffness during dynamic movements might indicate improved strategies for regulating joint stiffness, as the surrounding muscles can eccentrically absorb external forces over time while effectively protecting the articular structures [30,31]. Before the cognitive training, the ACLR in the present study showed greater mid-range stiffness in response to both fearful and specific sports-injury-related pictures than neutral pictures. However, there was not a significant difference in mid-range stiffness values between picture types after the cognitive training. This means that ACLR patients were negatively affected by fearful stimuli before the cognitive training, but at the end of the study, these patients stiffened their knees in response to adverse visual stimuli as well as when they viewed neutral pictures. This substantiates multiple prior findings [24,29,30], suggesting that the training of executive function skills has the potential to enhance preparatory and reactive dynamic restraint mechanisms subsequent to ACL injuries.

When individuals undergo ACLR, they often experience neuroplasticity, which can disrupt the cognitive processes necessary for appropriate motor coordination. ACLR patients may encounter challenges in areas such as attention, working memory, and motor planning, which can impact their ability to effectively coordinate movements and adapt to dynamic situations [32]. Therefore, cognitive training interventions may enhance cognitive sensory integration, facilitating the integration of proprioceptive and visual feedback within the CNS [33]. This improved integration can allow for a more precise interpretation of sensory information related to force and joint position, leading to optimal regulation of joint stiffness during movement. Further, improved cognitive sensory integration may enhance motor planning and coordination, particularly in response to unanticipated perturbation during physical activity [34]. Therefore, the improved joint stiffness regulation strategy is likely achieved by improving cognitive sensory integration and motor planning within the CNS.

Regardless of the cognitive training, the ACLR group showed poor self-reported knee function and significant differences for the single-legged hop for distance between limbs when compared to the CONT group. Nevertheless, following the cognitive training, the ACLR group exhibited substantial enhancements in knee functional performance within the reconstructed limb. Despite the absence of studies investigating the direct impact of cognitive training on knee function recovery after ACLR, there is an indication that executive function training can contribute to enhancing motor control [7]. One recent study implied that computerized speed-of-processing training can improve cognitive deficits following a traumatic brain injury (TBI), presumably through reinforcement of neural adaptation in the working memory network [35]. Our data may indicate that improvement of executive function facilitates cognitive processing related to motor coordination. Furthermore, our findings with 95% CI of mean differences with effect sizes for executive function assessments, joint stiffness, and knee function outcomes indicate the need for clinicians to consider neurocognitive factors as an intervention approach during rehabilitation in ACLR patients.

Our study highlights the potential benefits of cognitive training programs, specifically targeting executive function skills, in individuals with ACL injury. Integrating online brain exercise programs or other cognitive training interventions into traditional ACL rehabilitation protocols may enhance neuromuscular control and functional joint stability, particularly when facing unexpected threats to the knee joint during high-velocity physical maneuvers. Additionally, clinicians should be aware of the potential effects of emotional responses on muscle tone and neuromechanical coupling. Rehabilitation programs may need to address emotional regulation strategies to optimize neuromuscular control and maintain functional joint stability.

The present study has several limitations worth considering. Firstly, our assessment of executive function progress after the cognitive training program relied on the NIH-TB executive function assessment, which includes the DCCS and FICA tests. While these tests exhibit high reliability (DCCS: 0.94, FICA: 0.96) when compared to traditional pen-and-pencil tests [16], their primary focus revolves around evaluating attention and cognitive identification of specific objects among various visual cues. However, it is important to acknowledge that maximizing dynamic restraint mechanisms through optimized cognitive management skills may also rely on other aspects of executive function, including processing speed, working memory, and intelligence. Furthermore, executive function is highly associated with age and education level, but we only compared the overall computed scores. Although there was no significant age difference between groups, education levels varied between and within each group. Second, these executive functioning skills may not result in permanent neuroplasticity in the brain, which may continuously alter the existing cognitive neural network. Although cognitive training could maintain executive function associated with activities of daily living in older populations over time [36], it is unclear whether similar outcomes would appear in physically active populations. Third, the absence of a control group without cognitive training limited the determination of a direct causal relationship between cognitive training and the observed improvements in executive function, joint stiffness regulation, and knee function outcomes. Future studies should include ACLR patients without cognitive training, which will serve as a true control group that would provide the cognitive training effects on joint stiffness regulation strategy and knee function. Lastly, our ACLR group showed better cognitive function on the DCCS than the CONT group regardless of the online brain training. It is possible that improvements in joint stiffness regulation strategy and knee function in the ACLR group in the present study may be not observed in other ACLR patients with poor cognitive function skills.

Future studies may consider how these potential covariant factors influence the effects of cognitive training on the progress of executive function, joint stiffness regulation strategies, and knee function following ACL injuries. Randomized control design of longitudinal prospective cohort studies that incorporate cognitive training with traditional neuromuscular control rehabilitation exercise programs will allow for the determination of the effectiveness of cognitive training on muscle coordination as well as the functional outcomes of each patient. Simultaneous observation of brain activity using neuroimaging techniques such as functional MRI and/or electroencephalography will also provide insight into the neuromechanical links between cognition and neuromuscular control strategies.

## 5. Conclusions

This study investigated the impact of cognitive training on dynamic restraint mechanisms by measuring joint stiffness regulation and knee function outcomes in ACLR patients compared to healthy controls. The findings reveal that the ACLR group demonstrated improvements in executive function, joint stiffness regulation in response to fearful and injury-related stimuli, and knee function outcomes. These results indicate that cognitive training interventions have the potential to enhance cognitive processing, leading to the recovery of normalized stiffness regulation and improved knee function in ACLR patients. Further research could explore the combined effects of cognitive training and neuromuscular control rehabilitation programs on the dynamic restrain system.

## Figures and Tables

**Figure 1 healthcare-11-01875-f001:**
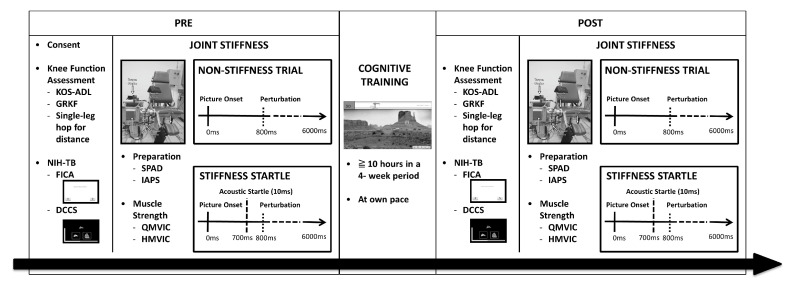
Research setup.

**Figure 2 healthcare-11-01875-f002:**
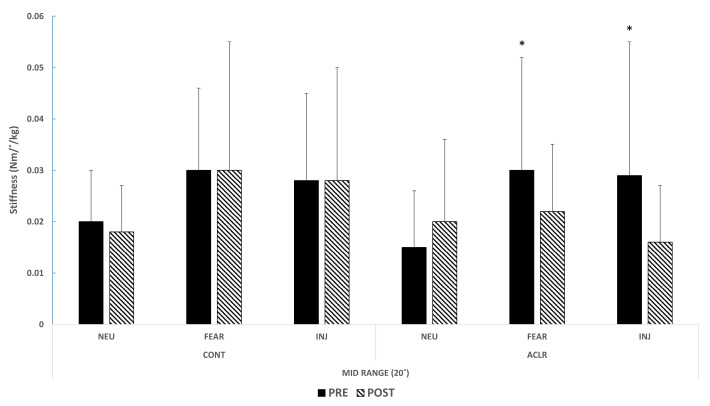
Pre–post cognitive training on mid-range stiffness between groups.

**Table 1 healthcare-11-01875-t001:** Participant demographic data.

	Demographic Data (Mean ± SD)		*p*-Value *
	CONT (*N* = 17)	ACLR (*N* = 16)	
Sex, n			
Male	4	4	
Female	13	12	
Age, yr	24.47 ± 4.97	22.19 ± 3.89	0.151
Height, cm	166.00 ± 8.80	166.21 ± 10.77	0.950
Weight, kg	62.51 ± 12.64	74.38 ± 27.53	0.130
Time from surgery, yr	…	3.9 ± 2.2	…

CONT—healthy controls; ACLR—ACL reconstructed patients. * Reported from t tests comparing group means.

**Table 2 healthcare-11-01875-t002:** NIH executive function and knee function outcomes.

		CONT		ACLR	
		PRE (Mean ± SD)	POST (Mean ± SD)	PRE (Mean ± SD)	POST (Mean ± SD)
NIH-TB	DCCS	9.44 ± 0.49	9.59 ± 0.47	9.80 ± 0.22 ^†^	9.83 ± 0.20 ^†^
	FICA	9.57 ± 0.32	9.72 ± 0.24 *	9.60 ± 0.27	9.71 ± 0.22 *
Knee function outcomes	GRFK (%)	100 ± 0	100 ± 0	95.12 ± 7.96 ^†^	94.19 ± 8.42 ^†^
	KOS-ADL (%)	100 ± 0	100 ± 0	93.05 ± 9.66 ^†^	95.81 ± 5.75 ^†^
	LSI (%)	100.61 ± 4.49	100.42 ± 2.17	93.44 ± 6.49 ^†^	96.49 ± 5.27 *^,†^

CONT—healthy controls; ACLR—ACL reconstructed patients; NIH-TB—NIH Toolbox for executive function assessment; PRE—before the cognitive training; POST—after the cognitive training; DCCS—dimensional change card sort test; FICA—flanker inhibitory control and attention test; GRKF—global rating of knee function; KOS-ADL—Knee Outcome Survey–Activities of Daily Living; LSI—hop limb symmetry index (% of involved limb’s distance to non-involved limb’s distance). * Significant PRE–POST time differences (*p* < 0.05). ^†^ Significant group differences (*p* < 0.05).

**Table 3 healthcare-11-01875-t003:** Joint stiffness values.

		Normalized Stiffness (Nm/°/kg) (Mean ± SD)			
		CONT		ACLR	
		PRE	POST	PRE	POST
Short-range (0–4°)	NEU	0.053 ± 0.007	0.052 ± 0.009	0.051 ± 0.016	0.051 ± 0.007
	FEAR	0.053 ± 0.009	0.055 ± 0.011	0.046 ± 0.012 ^†^	0.049 ± 0.007 ^†^
	INJ	0.059 ± 0.012	0.054 ± 0.012	0.047 ± 0.012 ^†^	0.047 ± 0.008 ^†^
Mid-range (0–20°)	NEU	0.020 ± 0.010	0.019 ± 0.009	0.016 ± 0.013	0.018 ± 0.016
	FEAR	0.031 ± 0.020	0.027 ± 0.019	0.028 ± 0.021 *	0.022 ± 0.013
	INJ	0.027 ± 0.016	0.023 ± 0.013	0.031 ± 0.025 *	0.017 ± 0.011
Long-range (0–40°)	NEU	0.048 ± 0.009	0.052 ± 0.014	0.040 ± 0.016	0.050 ± 0.013
	FEAR	0.047 ± 0.016	0.054 ± 0.013	0.050 ± 0.015	0.049 ± 0.018
	INJ	0.046 ± 0.014	0.054 ± 0.018	0.046 ± 0.018	0.052 ± 0.015

CONT—healthy controls; ACLR—ACL reconstructed patients; NEU—neutral pictures; FEAR—fearful pictures; INJ—sports knee injury-related pictures; PRE—before the cognitive training; POST—after the cognitive training. * Significant differences from NEU (*p* < 0.05). ^†^ Significant group differences (*p* < 0.05).

## Data Availability

To maintain confidentiality, the research data for the current study will be made available only upon request.

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
