# Peer review of "Cognitive Training Improves Joint Stiffness Regulation and Function in ACLR Patients Compared to Healthy Controls"

_healthcare, 2023, doi:10.3390/healthcare11131875_

Round 1

Reviewer 1 Report

I would like to congratulate the authors for the study. My comments are:

  • The introduction is short. Authors should include a greater number of references.
  • Lines 89-97: I don't understand it. In part of figure 1 or is it text? Authors should clarify it.
  • The first paragraph of the discussion should be deleted. The authors repeat content. They should start with the main finding of their study.
  • Authors should include a "Practical Applications" section before the limitations paragraph.
  • The conclusion is very long. It should be a paragraph with 2-3 sentences explaining the main finding of your study
  • The reference style is correct.

Author Response

  • The introduction is short. Authors should include a greater number of references.

Reply: We appreciate your feedback on our manuscript. We have included additional references in the revised version of the introduction section to provide a more comprehensive overview of the topic.

  • Lines 89-97: I don't understand it. In part of figure 1 or is it text? Authors should clarify it.

Reply: Yes, it is the figure 1 legend. We apologize for any confusion caused by the lack of clarity in the manuscript. To address this issue, we changed its font size to 9 (line 90-97).

  • The first paragraph of the discussion should be deleted. The authors repeat content. They should start with the main finding of their study.

Reply: Thank you for your comment. You have deleted the first paragraph of the discussion.

  • Authors should include a "Practical Applications" section before the limitations paragraph.

Reply: Thank you for your insightful comment. In our revised manuscript, we have made efforts to include clinical implication in the discussion (line 368-376)

  • The conclusion is very long. It should be a paragraph with 2-3 sentences explaining the main finding of your study

Reply: Thank you for your comment. We have taken your suggestion into consideration and revised the conclusion to provide a more concise summary of the main findings (line 414-422).

  • The reference style is correct.

Reviewer 2 Report

This study seems to have the aim of finding out whether cognitive training is effective for recovery after ACLR, but the study design is not in line with RCTs: instead of comparing an intervention group with cognitive training and a control group without cognitive training, healthy subjects were used as controls to see whether cognitive training was more effective for ACLR patients or not. It is difficult to see what significance this would have even if significant differences could be demonstrated as in the results.

 I am afraid that whether or not cognitive training is effective for healthy subjects is a question that most readers are not interested in, and as there is no adequate control, this study failed to demonstrate whether or not cognitive training is effective for ACLR. 

At the very least, the authors must clearly state the reasons why they did not include a group without training as a control in the study design and why they considered it important to compare healthy subjects and patients with ACLR.

 As minor points, gender is an important confounding factor in this type of research, so, if possible, analyses should be conducted separately for men and women. Furthermore, it is worrying that the mean weight of the control group is much higher than that of the sick group. This is not a significant difference due to the greater variance in the ACLR group and indicates the existence of clearly emaciated persons in the ACLR group, but the mean weight of the control group should have been at least a little lower.

Author Response

This study seems to have the aim of finding out whether cognitive training is effective for recovery after ACLR, but the study design is not in line with RCTs: instead of comparing an intervention group with cognitive training and a control group without cognitive training, healthy subjects were used as controls to see whether cognitive training was more effective for ACLR patients or not. It is difficult to see what significance this would have even if significant differences could be demonstrated as in the results.

 Reply: Thank you for pointing out the limitations in our study design. We acknowledge that our study did not follow a traditional randomized controlled trial design, which typically involves comparing an intervention group receiving cognitive training with a control group without cognitive training. Instead, we used healthy subjects as controls to assess the relative effectiveness of cognitive training in ACLR patient. We understand that the absence of a dedicated ACLR control group receiving no cognitive training makes it challenging to determine the true efficacy of cognitive training in the recovery after ACLR. We appreciate your feedback and recognize the importance of future research employing a more rigorous RTC design to provide clearer evidence regarding the effectiveness of cognitive training in ACLR patients. Therefore, we have made effort to discuss its limitation in the discussion section (line 393-398).

 I am afraid that whether or not cognitive training is effective for healthy subjects is a question that most readers are not interested in, and as there is no adequate control, this study failed to demonstrate whether or not cognitive training is effective for ACLR. 

 Reply: Thank you for expressing your concerns. While it is true that the effectiveness of cognitive training in healthy subjects may not be of immediate interest to all readers, out study aimed to explore the potential benefits of cognitive training specifically in ACLR patients. By comparing the outcomes of ACLR patients who underwent cognitive training to those healthy controls, we sought to assess whether cognitive training had a differential impact on the recovery of ACLR patients compared to individuals without ACLR. While the absence of a dedicated control group without cognitive training in both ACLR and healthy groups is a limitation, we hope our findings sill provide valuable insights into the potential effectiveness of cognitive training in the context of ACLR rehabilitation. We have acknowledged this limitation in our discussion and emphasized the need for further research with a more robust study design.

At the very least, the authors must clearly state the reasons why they did not include a group without training as a control in the study design and why they considered it important to compare healthy subjects and patients with ACLR.

  Reply: Thank you for your valuable comment. As mentioned in the previous responses, we acknowledge the absence of a control group without cognitive training in our study design and the importance of including such a group for a more robust comparison, which would have strengthened the study, and we appreciate your suggestions.

 As minor points, gender is an important confounding factor in this type of research, so, if possible, analyses should be conducted separately for men and women. Furthermore, it is worrying that the mean weight of the control group is much higher than that of the sick group. This is not a significant difference due to the greater variance in the ACLR group and indicates the existence of clearly emaciated persons in the ACLR group, but the mean weight of the control group should have been at least a little lower.

Reply: Thank you for your comment. We acknowledge the importance of considering gender as a potential confounding factor in this type of research. However, due to the limited sample size in our study, conducting separate gender analyses was not feasible. Nonetheless, we took gender into account during the matching processing between the ACLR and control groups. Regarding the difference in mean weight between the groups, we acknowledge the concern. Despite the statistically non-significant difference in mean weight, it is important to note that the matching process aimed to ensure similar demographic characteristics between the groups, including body weight. We will take considerations into account in our future research to provide more comprehensive insights into the effect of cognitive training on ACLR patient.

Reviewer 3 Report

It is an interesting manuscript. However, I have a few comments.

The introduction should be improved. It should include studies previously done on this topic.

Line 89-96- I didn’t understand this. Is it the figure legend?

The calculated sample size was 12 in each group. Why are there more participants recruited and assessed? This may show a high significance in hypothesis testing.

The result and discussion are satisfactory   

Author Response

The introduction should be improved. It should include studies previously done on this topic.

Reply: We appreciate your feedback on the inclusion of previous studies in our manuscript. We have taken note of your suggestion to include studies previously conducted in the introduction. However, we would like to clarify that due to the unique nature of our research focus, there have been no prior studies specifically examining the effects of cognitive training on neuromuscular control in ACLR patients. Instead, you have added additional references that support the background and rationale for our study.

Line 89-96- I didn’t understand this. Is it the figure legend?

Reply: Yes, it is the figure 1 legend. We apologize for any confusion caused by the lack of clarity in the manuscript. To address this issue, we changed its font size to 9 (line 90-97).

The calculated sample size was 12 in each group. Why are there more participants recruited and assessed? This may show a high significance in hypothesis testing.

Replay: As part of our study planning, we incorporated a dropout rate of up to 40% into our sample size calculation because participants were completely voluntarily participated for two testing session along with 4 weeks of cognitive training. This was done to account for potential attrition during the course of the study, ensuring that we would still have an adequate number of participants to maintain statistical power and draw meaningful conclusion. However, we were fortunate to have a lower dropout rate than initially expected, which has provided us with a larger final sample size than originally planned. While the lower dropout rate may affect the statistical power of our analysis, it is important to note that having a larger sample size can still provide valuable insights and enhance the reliability of the study findings. The increased number of participants allows for more robust statistical analysis, strengthens the generalizability of the results, and provides a more comprehensive understanding of the research question at hand.

Reviewer 4 Report

Strengths: The results showed that cognitive training can affect functional improvement in ACLR patients, so it will be more effective when performed together with exercise.

Weaknesses: It seems necessary to compare the patient group, not the normal group.

1. Please fill out the reliability and validity of the evaluation tool.

2. Please also include inclusion criteria.

3. Wouldn't .25 for an effect size be a low value? Please also write which variable you used.

4. Please describe the structure of The Brain Training for High Achievers course from the BrainHQ (Posit Science Corp. San Francisco, CA) in detail.

5. Is there no normality test?

6. Further elucidation of the mechanism of how cognitive training affects patients with  ACLR seems to be necessary. It is necessary to consider the overall effect of cognitive training being more effective for ACLR patients.

7. Please check the spelling and grammar of the text.

8. Please add more clinical significance to the discussion

Author Response

1. Please fill out the reliability and validity of the evaluation tool.

Reply: Thank you for your comment. We would like to acknowledge that we have included the reliability of the evaluation tool with references in the discussion section, particularly at line 380 in the revised version of our manuscript.

2. Please also include inclusion criteria.

Reply: Thank you for your comment. We have inclusion criteria, lines 78-83 in the revised version of our manuscript.

3. Wouldn't .25 for an effect size be a low value? Please also write which variable you used.

Reply: Thank you for bring up the concern regarding the effect size value of 0.25 mentioned in our study. The chosen effect size is typically used during the sample size calculation to estimate the number of participants needed to detect the expected effects. It does not directly reflect the clinical or practical significant of the observed results.

4. Please describe the structure of The Brain Training for High Achievers course from the BrainHQ (Posit Science Corp. San Francisco, CA) in detail.

Reply: Thank you for your valuable comment. We have addressed details of the Brain Training for High Achievers course in the method section (line 112-120).

5. Is there no normality test?

Reply: I apologize for the oversight. Yes, a normality test was conducted to assess the distribution of the data. The Shapiro-Wilk test was used to examine the normality assumption and it has been added to the revised manuscript (line 188-189). Thank you for pointing out the omission, and I appreciate your attention to detail.

6. Further elucidation of the mechanism of how cognitive training affects patients with  ACLR seems to be necessary. It is necessary to consider the overall effect of cognitive training being more effective for ACLR patients.

Reply: Thank you for your comment. In our revised manuscript, we have made efforts to explore and discuss possible mechanisms of cognitive training impact on neuromuscular control in ACLR patients (line 340-352)

7. Please check the spelling and grammar of the text.

Reply: We appreciate your feedback regarding the spelling and grammar of the text. The revised manuscript has undergone two rounds of professional proofreading to ensure accurate spelling and grammar.

8. Please add more clinical significance to the discussion

Reply: Thank you for your comment. In our revised manuscript, we have made efforts to include clinical implication in the discussion (line 368-376)

Round 2

Reviewer 2 Report

Although the lack of the control may be a disadvantage of this study, it is understandable that there are unavoidable limitations of the study plan in the practice of medicine, and the authors have included discussion of this in the revised manuscript and the results appear to contain useful information for readers, I agree to accept this article for publication.

Minor points

The pre and post in Figure 1 (search setting) are identical and need not be repeated in detail (rather confusing trying to find the differences).

Reviewer 3 Report

The paper is improved now 

No further comments